# SLM Lab: A Comprehensive Benchmark and Modular Software Framework for Reproducible Deep Reinforcement Learning

## Abstract

We introduce SLM Lab, a software framework for reproducible reinforcement learning (RL) research. SLM Lab implements a number of popular RL algorithms, provides synchronous and asynchronous parallel experiment execution, hyperparameter search, and result analysis. RL algorithms in SLM Lab are implemented in a modular way such that differences in algorithm performance can be confidently ascribed to differences between algorithms, not between implementations. In this work we present the design choices behind SLM Lab and use it to produce a comprehensive single-codebase RL algorithm benchmark. In addition, as a consequence of SLM Lab's modular design, we introduce and evaluate a discrete-action variant of the Soft Actor-Critic algorithm (Haarnoja et al., 2018) and a hybrid synchronous/asynchronous training method for RL agents.

## 1 Introduction

Progress in reinforcement learning (RL) research proceeds only as quickly as researchers can implement new algorithms and publish reproducible empirical results. But it is no secret that modern RL algorithms are hard to implement correctly (Tucker et al., 2018), and many empirical results are challenging to reproduce (Henderson et al., 2017; Islam et al., 2017; Machado et al., 2017). Addressing these problems is aided by providing better software tools to the RL research community.

In this work we introduce SLM Lab,[1] a software framework for reinforcement learning research designed for reproducibility and extensibility. SLM Lab is the first open source library that includes algorithm implementations, parallelization, hyperparameter search, and experiment analysis in one framework.

After presenting the design and organization of SLM Lab, we demonstrate the correctness of its implementations by producing a comprehensive performance benchmark of RL algorithms across 77 environments. To our knowledge this is the largest single-codebase RL algorithm comparison in the literature. In addition, we leverage the modular design of SLM Lab to introduce a variant of the SAC algorithm for use in discrete-action-space environments and a hybrid synchronous/asynchronous training scheme for RL algorithms.

## 2 SLM Lab

### 2.1 Library Organization

Modularity is the central design choice in SLM Lab, as depicted in Figure 1. Reinforcement learning algorithms in SLM Lab are built around three base classes:

- `Algorithm`: Handles interaction with the environment, implements an action policy, computes the algorithm-specific loss functions, and runs the training step.
- `Net`: Implements the deep networks that serve as the function approximators for an `Algorithm`.

---

[1]URL redacted to preserve anonymity.

- `Memory`: Provides the data storage and retrieval necessary for training.

The deep learning components in SLM Lab are implemented using PyTorch Paszke et al. (2017). The `Net` and `Memory` classes abstract network training details, data storage, and data retrieval, simplifying algorithm implementations. Furthermore, many `Algorithm` classes are natural extensions of each other.

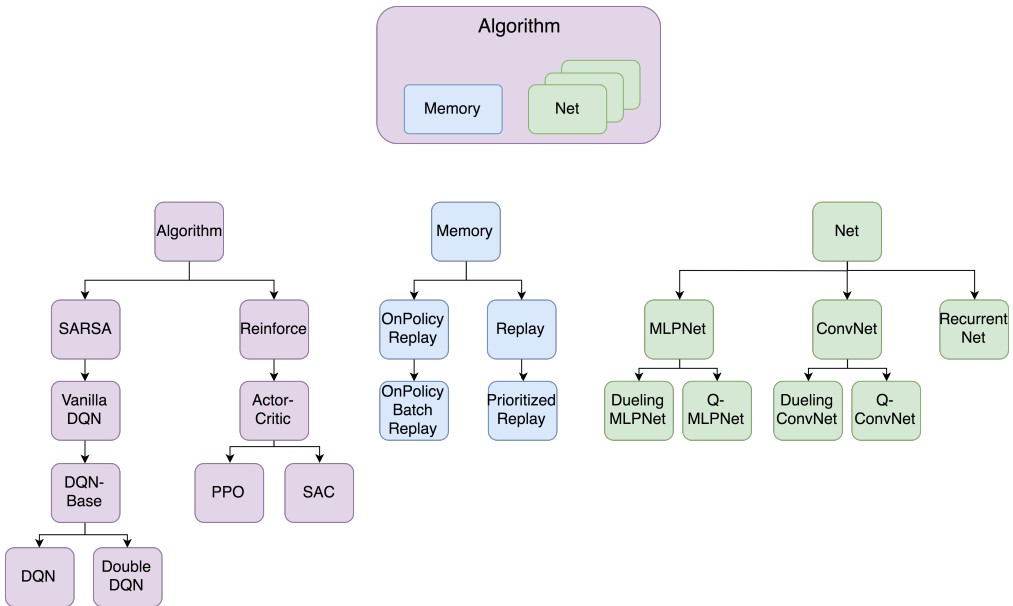

Figure 1: SLM Lab classes and their inheritance structure.

Modular code is critical for deep RL research because many RL algorithms are extensions of other RL algorithms. If two RL algorithms differ in only a small way, but a researcher compares their performance by running a standalone implementation of each algorithm, they cannot know whether differences in algorithm performance are due to meaningful differences between the algorithms or merely due to quirks in the two implementations. Henderson et al. (2017) showcase this, demonstrating significant performance differences between different implementations of the same algorithm.

Modular code is also important for research progress. It makes it as simple as possible for a researcher to implement — and reliably evaluate — new RL algorithms. And for the student of RL, modular code is easier to read and learn from due to its brevity and organization into digestible components.

Proximal Policy Optimization (PPO) (Schulman et al., 2017) is a good example. When considered as a stand alone algorithm, PPO has a number of different components. However, it differs from the Actor-Critic algorithm only in how it computes the policy loss, runs the training loop, and by needing to maintain an additional actor network during training. Figure 2 shows how this similarity is reflected in the SLM Lab implementation of PPO.

The result is that the `PPO` class in SLM Lab has five overridden methods and contains only about 140 lines of code. Implementing it was straightforward once the `ActorCritic` class was implemented and thoroughly tested. More importantly, we can be sure that the performance difference between Actor-Critic and PPO observed in experiments using SLM Lab, shown below in Section 3, are due to something in the 140 lines of code that differentiate `ActorCritic` and `PPO`, and not to other implementation differences.

Another example of modularity in SLM Lab is that, thanks to the consistent API shared between `Algorithm`, `Net`, and `Memory` subclasses, synchronous parallelization using vector environments (Dhariwal et al., 2017) can be combined with asynchronous parallelization of an individual RL agent's learning algorithm. This multi-level parallelization is further discussed in Section 3.3, where we demonstrate its performance benefits.

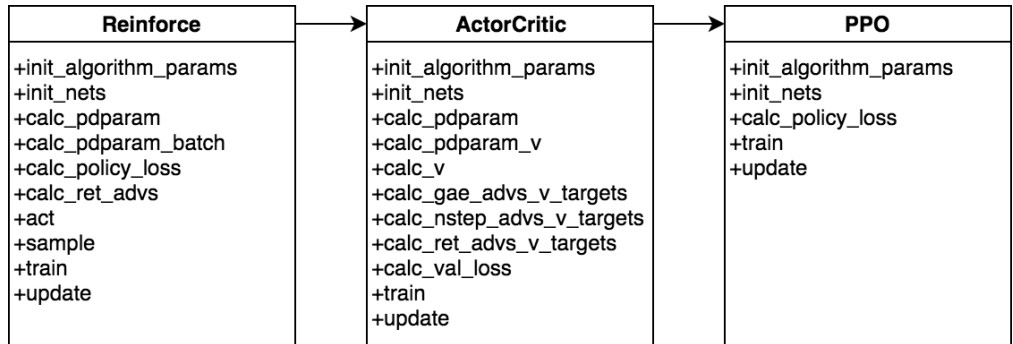

Figure 2: `Reinforce`, `ActorCritic`, and `PPO` class methods in SLM Lab. + indicates that a method is added or overridden in the class.

## 2.2 EXPERIMENT ORGANIZATION

Reinforcement learning algorithms vary greatly in their performance across different environments, hyperparameter settings, and even within a single environment due to inherent randomness. SLM Lab is designed to easily allow users to study all these types of variability.

SLM Lab organizes experiments in the following hierarchy, shown in Figure 3:

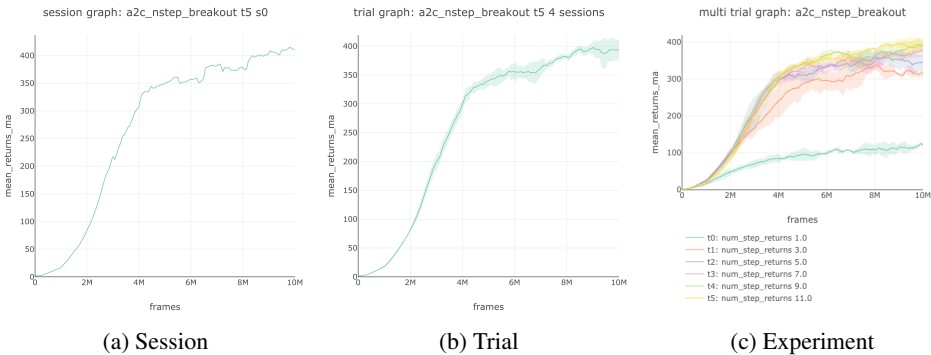

(a) Session        (b) Trial        (c) Experiment

Figure 3: Experiment organization in SLM Lab. A Session is a single run of an algorithm on an environment. A Trial is a collection of Sessions. An Experiment is a collection of Trials with different algorithms and/or environments.

1. **Session** The lowest level of the SLM Lab experiment framework: a single training run of one agent on one environment with one set of hyperparameters, all with a fixed random seed.

2. **Trial** A trial consists of multiple Sessions, with the Sessions varying only in the random seed.

3. **Experiment** Generates different sets of hyperparameters (according to a spec file, see below) and runs a Trial for each one. It can be thought of as a study, e.g. "What values of $n$ of A2C n-step returns provide the fastest, most stable solution, if the other variables are held constant?"

SLM Lab automatically produces plots for Sessions, Trials, and Experiments for any combination of environments and algorithms. It also logs and tracks metrics during training such as rewards, loss, exploration and entropy variables, model weights and biases, action distributions, frames-per-second and wall-clock time. The metrics are also visualized using TensorBoard (Abadi et al., 2015). Hyperparameter search is implemented using Ray Tune (Liaw et al., 2018), and the results are automatically analyzed and presented hierarchically in increasingly granular detail.

## 2.3 REPRODUCIBILITY

The complexity of RL algorithms makes reproducing RL results challenging (Henderson et al., 2017; Machado et al., 2017). Every RL researcher knows the difficulty of trying to reproduce an algorithm from its description in a paper alone. Even if code is published along with RL research results, the key algorithm design choices (or mistakes (Tucker et al., 2018)) are often buried in the algorithm implementation and not exposed naturally to the user.

In SLM Lab, every configurable hyperparameter for an algorithm is specified in a *spec file*. The spec file is a JSON file containing a git SHA and all the information required to reproduce a Session, Trial, or Experiment as per Section 2.2. Reproducing the entirety of an RL experiment merely requires checking out the code at the git SHA and running the saved spec file.

The main entries in a spec file are given below. Examples of spec files are given in full in Supplementary Section A.4.

1. **agent -** A list (to allow for multi-agent settings in the future), each element of which contains the spec for a single agent. Each agent spec contains the details for its components as described in Section 2.1:

    (a) **algorithm.** The main parameters specific to the algorithm, such as the policy type, exploration strategy (e.g. $\epsilon$-greedy, Boltzmann), algorithm coefficients, rate decays, and training schedules.

    (b) **memory.** Specifies which memory to use as appropriate to the algorithm along with any specific memory hyperparameters such as the batch size and the memory size.

    (c) **net.** The type of neural network, its hidden layer architecture, activations, gradient clipping, loss function, optimizer, rate decays, update method, and CUDA usage.

2. **env -** A list (to allow for multi-environment settings in the future), each element of which specifies an environment. Each environment spec includes an optional maximum time step per episode, the total time steps (frames) in a Session, the state and reward preprocessing methods, and the number of environments in a vector environment.

3. **body -** Specifies how (multi-)agents connect to (multi-)environments.

4. **meta -** The high-level configuration of how the experiment is to be run. It gives the number of Trials and Sessions to run, the evaluation and logging frequency, and a toggle to activate asynchronous training.

5. **search -** The hyperparameters to search over and the methods used to sample them. Any variables in the spec file can be searched over, including environment variables.

## 3 RESULTS

Reporting benchmark results is essential for validating algorithm implementations. SLM Lab maintains a benchmark page and a public directory containing all of the experiment data, models, plots, and spec files associated with the reported results.[2] We welcome contributions to this benchmark page via a Pull Request.

We tested the algorithms implemented in SLM Lab on 77 environments: 62 Atari games and 11 Roboschool environments available through OpenAI gym (Brockman et al., 2016) and 4 Unity environments (Juliani et al., 2018). These environments span discrete (Table 1) and continuous (Table 2) control problems with high- and low-dimensional state spaces.

The results we report in each of the tables are the score per episode at the end of training averaged over the previous 100 training checkpoints. Agents were checkpointed every 10k (Atari) or 1k (Roboschool, Unity) training frames. This measure is less sensitive to rapid increases or decreases in performance, instead reflecting average performance over a substantial number of training frames.

To our knowledge, the results we present below are a more comprehensive performance comparison than has been previously published for a single codebase. A full set of learning curves as well as a full table of results for the Atari environments are provided in the supplementary materials.

---

[2]URL redacted to preserve anonymity.

## 3.1 EXPERIMENT DETAILS

A complete set of spec files listing all hyperparameters for all algorithms and experiments are included in the `slm_lab/spec/benchmark` directory of SLM Lab as well in the experiment data released along with the results. Example spec files listing all of the hyperparameters for PPO and DDQN + PER on the Atari and Roboschool environments are included in Supplementary Section A.4.

For the Atari environments, agents were trained for 10M frames (40M accounting for skipped frames). For the Roboschool environments, agents were trained for 2M frames except for RoboschoolHumanoid (50M frames), RoboschoolHumanoidFlagrun (100M) and RoboschoolHumanoidFlagrunHarder (100M). For the Unity environments, agents were trained for 2M frames. Training was parallelized either synchronously or with a hybrid of synchronous and asynchronous methods.

Our results for PPO and A2C on Atari games are comparable the results published by Schulman et al. (2017). The results on DQN and DDQN + PER on Atari games are mixed: at the same number of training frames[3] we sometimes match or exceed the reported results and sometimes perform worse. This is likely due to two hyperparameter differences. We used a replay memory of size 200,000 compared to 1M in Mnih et al. (2015), van Hasselt et al. (2015), and Schaul et al. (2015). The final output layer of the network is smaller fully-connected layer with 256 instead of 512 units. Finally, our results for SAC confirm the strength of this algorithm compared to PPO for continuous control problems. However the absolute performance is typically worse than the published results from Haarnoja et al. (2018b). Due to computational constraints, SAC was trained with a replay buffer of 0.2M elements and combined experience replay (Zhang & Sutton, 2017) compared with 1M elements in Haarnoja et al. (2018b) and this is potentially a significant difference.

Table 1: Episode score at the end of training attained by SLM Lab implementations on discrete-action control problems. Reported episode scores are the average over the last 100 checkpoints, and then averaged over 4 Sessions. A Random baseline with score averaged over 100 episodes is included. Results marked with '*' were trained using the hybrid synchronous/asynchronous version of SAC to parallelize and speed up training time. For SAC, Breakout, Pong and Seaquest were trained for 2M frames instead of 10M frames.

| Environment | Algorithm | | | | | | |
| --- | --- | --- | --- | --- | --- | --- | --- |
| | Random | DQN | DDQN+PER | A2C (GAE) | A2C (n-step) | PPO | SAC |
| Breakout | 1.26 | 80.88 | 182 | 377 | 398 | **443** | 3.51* |
| Pong | -20.4 | 18.48 | 20.5 | 19.31 | 19.56 | **20.58** | 19.87* |
| Seaquest | 106 | 1185 | **4405** | 1070 | 1684 | 1715 | 171* |
| Qbert | 157 | 5494 | 11426 | 12405 | **13590** | 13460 | 923* |
| LunarLander | -162 | 192 | 233 | 25.21 | 68.23 | 214 | **276** |
| UnityHallway | -0.99 | -0.32 | 0.27 | 0.08 | -0.96 | **0.73** | 0.01 |
| UnityPushBlock | -1.00 | 4.88 | 4.93 | 4.68 | 4.93 | **4.97** | -0.70 |

## 3.2 SOFT ACTOR-CRITIC FOR DISCRETE ENVIRONMENTS

All published results for the Soft Actor-Critic (SAC) algorithm (Haarnoja et al., 2018a;b) are for continuous control environments. However, nothing in its algorithmic description makes it unsuitable in principle for use in discrete action-space environments. As a consequence of the modular structure of SLM Lab, it was straightforward to design and implement a discrete variant of SAC. We did so by using policy `Nets` that produced parameters of a Gumbel-Softmax distribution (Jang et al., 2016; Maddison et al., 2016) from which discrete actions were sampled. The results of this discrete variant of SAC are in Table 1.

On environments for which its training converged, we found SAC to be of comparable or better sample efficiency than all of the other algorithms that we tested. For example, SAC achieves an

---

[3]Estimated using the training curves provided in Figure 7 of Schaul et al. (2015)

Table 2: Episode score at the end of training attained by SLM Lab implementations on continuous control problems. Reported episode scores are the average over the last 100 checkpoints, and then averaged over 4 Sessions. A Random baseline with score averaged over 100 episodes is included. Results marked with '*' require 50M-100M frames, so we use the hybrid synchronous/asynchronous version of SAC to parallelize and speed up training time.

| Environment | Algorithm | | | | |
|---|---|---|---|---|---|
| | Random | A2C (GAE) | A2C (n-step) | PPO | SAC |
| RoboschoolAnt | 52.30 | 787 | 1396 | 1843 | **2915** |
| RoboschoolAtlasForwardWalk | 40.23 | 59.87 | 88.04 | 172 | **800** |
| RoboschoolHalfCheetah | 0.83 | 712 | 439 | 1960 | **2497** |
| RoboschoolHopper | 21.22 | 710 | 285 | 2042 | **2045** |
| RoboschoolInvertedDoublePendulum | 285 | 996 | 4410 | 8076 | **8085** |
| RoboschoolInvertedPendulum | 23.5 | **995** | 978 | 986 | 941 |
| RoboschoolReacher | -9.24 | 12.9 | 10.16 | 19.51 | **19.99** |
| RoboschoolWalker2d | 16.06 | 280 | 220 | 1660 | **1894** |
| RoboschoolHumanoid | -36.16 | 99.31 | 54.58 | 2388 | **2621*** |
| RoboschoolHumanoidFlagrun | -6.15 | 73.57 | 178 | 2014 | **2056*** |
| RoboschoolHumanoidFlagrunHarder | -8.64 | -429 | 253 | **680** | 280* |
| Unity3DBall | 1.24 | 33.48 | 53.46 | 78.24 | **98.44** |
| Unity3DBallHard | 1.17 | 62.92 | 71.92 | 91.41 | **97.06** |

average score over the previous 100 checkpoints of around 20 after 1M frames on Atari Pong, whereas all other algorithms that we tested require at least 2M frames to achieve the same result (plot shown in Supplementary Section A.3). However, even though SAC trained successfully on Pong and Lunar Lander, we were not able to successfully train it on all other Atari environments. We also note that while SAC is sample efficient it is more computationally expensive than the other algorithms, which presents an obstacle for extensive performance tuning.

## 3.3 Hybrid synchronous and asynchronous training

Synchronous and asynchronous parallelization can be combined in SLM Lab to accelerate training, as shown in Figure 4. SLM Lab implements synchronous parallelization within Sessions (Section 2.2) using vector environments (Dhariwal et al., 2017) and asynchronous parallelization within Trials (Section 2.2) using multiple workers, one per Session. There are two available methods for Trial level parallelization; Hogwild! (Recht et al., 2011) or a server-worker model in which workers periodically push gradients to a central network and pull copies of the updated parameters.

If training is constrained by data sampling from the environment, then increasing the number of vector environments (synchronous parallelization) speeds up training. But this speed-up saturates as the training step becomes a bottleneck and the environment waits for the agent to train. In the example shown in Figure 4 the frames per seconds (fps) increases from around 200 for a single worker and 1 environment to around 360 for 1 worker and 16 environments, and saturates thereafter. Once fps becomes constrained by the training step, it is beneficial to add workers (asynchronous parallelization) to effectively parallelize the parameter updates. A hybrid of 16 workers each with 4 environments resulted in the maximum fps of around 3800.

## 4 Related Work

### 4.1 Reproducibility in Reinforcement Learning

The instability of RL algorithms (Haarnoja et al., 2018a), randomness in agent policies and the environment (Henderson et al., 2017; Islam et al., 2017), as well as differences in hyperparameter tuning (Islam et al., 2017) and implementations (Henderson et al., 2017; Tucker et al., 2018) all contribute to the challenge of reproducing RL results.

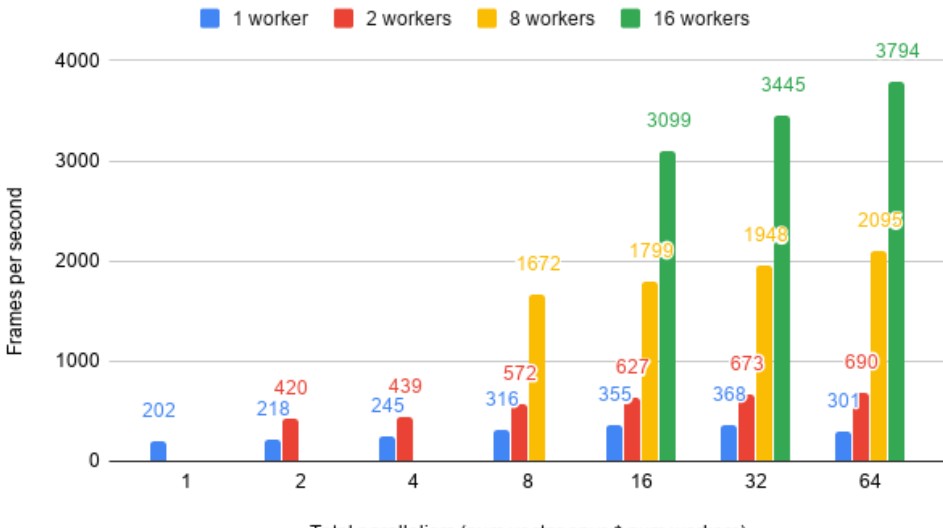

Figure 4: Average frames per second as number of vector environments and Hogwild! workers are varied. Each setting was trained using PPO with the same hyperparameters on the RoboschoolAnt environment.

Consequently, the importance of comprehensively documenting all hyperparameters along with published results and software is well recognized within the research community (Machado et al., 2017; Castro et al., 2018).

## 4.2 SOFTWARE FOR REINFORCEMENT LEARNING

To date more than twenty reinforcement-learning-themed open source software libraries have been released. They can be organized into two categories: those implementing RL algorithms and those implement RL environments.[4] SLM Lab is an algorithm-focused library with built-in integration with the OpenAI gym (Brockman et al., 2016), OpenAI Roboschool, VizDoom (Kempka et al., 2016), and Unity ML-Agents (Juliani et al., 2018) environment libraries.

Table 3 summarizes the algorithm-focused reinforcement learning software libraries.

Libraries such as Catalyst (Kolesnikov, 2018), ChainerRL (chainer, 2017; Tokui & Oono, 2015), coach (Caspi et al., 2017), DeepRL (Zhang, 2017), OpenAI baselines (Dhariwal et al., 2017), RL-graph (Schaarschmidt et al., 2019), RLkit (Pong, 2018), rlpyt (Stooke & Abbeel, 2019), RLLib (Liang et al., 2017), Stable Baselines (Hill et al., 2018), TensorForce (Kuhnle et al., 2017), TF-Agents (Guadarrama et al., 2018), and vel (Tworek, 2018) implement a wide variety of algorithms and are intended to be applied to a variety of RL problems. Most of these libraries also provide some benchmark results for the implemented algorithms to validate their performance. These can be thought of as *generalist* RL libraries and are the most closely related to SLM Lab.

Other libraries focus on specific algorithms (Dopamine (Castro et al., 2018), Softlearning (Haarnoja et al., 2018c), a2c-ppo-acktr-gail (Kostrikov, 2018), Keras-RL (Plappert, 2016)), problems (Open-Spiel (Lanctot et al., 2019), ELF (Tian et al., 2019), MAgent (Zheng et al., 2017), reaver (Ring, 2018)), components such as loss functions (TRFL (DeepMind, 2018) or scaling training (Horizon (Gauci et al., 2018)).

The use of configuration files to specify hyperparameters varies. Catalyst, coach, DeepRL, Dopamine, Horizon, reaver, Softlearning, RLgraph, RLLib, and vel use configuration files and provide a number of configured examples. In most cases the network architecture is excluded from the

---

[4]A few libraries such as OpenSpiel (Lanctot et al., 2019) and ELF (Tian et al., 2019) implement both.

Table 3: Comparison of RL software libraries. Algorithm acronyms are explained in Supplementary Section A.1. REINFORCE is excluded as are less well-known algorithms. "Benchmark" indicates whether the library reports the performance of their implementations. "Config" indicates whether hyperparameters are specified separately from the implementation and run scripts; "split" indicates that the configuration is divided across multiple files, "partial" indicates that some but not all hyperparameters are included. "Parallel" denotes whether training for any algorithms can be parallelized. "HPO" denotes support for hyperparameter optimization. "Plot" denotes whether the library provides any methods for visualizing results.

| Library | Algorithms | Benchmark | Config | Parallel | HPO | Plot |
|---|---|---|---|---|---|---|
| a2c-ppo-acktr-gail | A2C, ACKTR, GAIL, PPO | ✓ | ✗ | ✓ | ✗ | ✓ |
| Baselines | A2C, ACER, ACKTR, DDPG, DQN, GAIL, HER, PPO, TRPO | ✓ | ✗ | ✓ | ✗ | ✓ |
| Catalyst | CRA, TD3, SAC, DDPG, QR | ✓ | ✓ | ✓ | ✗ | ✓ |
| ChainerRL | A3C, ACER, C51, DDPG, DQN+, IQN, NSQ, PPO, Rainbow, TRPO, TD3, SAC | ✓ | ✗ | ✓ | ✗ | ✓ |
| coach | A3C, ACER, C51, DDPG, DQN+, QR-DQN, NAF, NEC, NSQ, PPO, Rainbow, SAC, TD3 | ✓ | split | ✓ | ✗ | ✓ |
| DeepRL | A2C, C51, DDPG, DQN+, NSQ, OC, PPO, QR-DQN, TD3 | ✓ | split | ✗ | ✗ | ✓ |
| Dopamine | DQN, C51, Rainbow | ✓ | split | ✓ | ✗ | ✓ |
| ELF | A3C, DQN, MCTS, TRPO | ✓ | ✗ | ✓ | ✗ | ✗ |
| Keras-RL | DDPG, DQN+, NAF, CEM, SARSA | ✗ | ✗ | ✓ | ✗ | ✓ |
| Horizon | C51, DQN+, SAC, TD3 | ✗ | ✓ | ✓ | ✗ | ✓ |
| MAgent | A2C, DQN | ✗ | ✗ | ✓ | ✗ | ✓ |
| OpenSpiel | A2C, DQN, MCTS | ✗ | ✗ | ✗ | ✗ | ✗ |
| pytorch-rl | A2C, DDPG, DQN+, HER | ✗ | ✗ | ✗ | ✗ | ✗ |
| reaver | A2C, PPO | ✓ | split | ✓ | ✗ | ✓ |
| RLgraph | A2C, Ape-X, DQN+, DQFD, IMPALA, PPO, SAC | ✓ | ✓ | ✓ | ✗ | ✗ |
| RLkit | DQN+, HER, SAC, TDM, TD3 | ✗ | ✗ | ✗ | ✗ | ✓ |
| RLLib | A3C, Ape-X, ARS, DDPG, DQN, ES, IMPALA, PPO, SAC | ✓ | partial | ✓ | ✗ | ✗ |
| rlpyt | A2C, DDPG, DQN+, CAT-DQN, PPO, TD3, SAC | ✓ | ✗ | ✓ | ✗ | ✗ |
| Softlearning | SAC | ✗ | ✓ | ✓ | ✗ | ✗ |
| Stable Baselines | A2C, ACER, ACKTR, DDPG, DQN, GAIL, HER, PPO, SAC, TD3, TRPO | ✓ | ✗ | ✓ | ✓ | ✓ |
| TensorForce | A3C, DQN+, DQFD, NAF, PPO, TRPO | ✗ | ✗ | ✓ | ✓ | ✗ |
| TF-Agents | DDPG, DQN+, PPO, SAC, TD3 | ✗ | ✗ | ✗ | ✗ | ✓ |
| TRFL | - | ✗ | ✗ | ✗ | ✗ | ✗ |
| vel | A2C, ACER, DDPG, DQN+, Rainbow, PPO, TRPO | ✓ | split | ✓ | ✗ | ✗ |
| SLM Lab | A2C, A3C, CER, DQN+, PPO, SAC | ✓ | ✓ | ✓ | ✓ | ✓ |

main configuration file and specified elsewhere. Catalyst, Horizon and RLgraph include all hyperparameters in a single configuration file and are the most similar to SLM Lab. However, RLgraph does

not include environment information, and none use the same file for configuring hyperparameter search.

Almost all libraries include some methods for parallelizing agent training, especially for on-policy methods. However most do not include hyperparameter optimization as a feature. The two exceptions are Stable Baselines (Hill et al., 2018) and Tensorforce (Kuhnle et al., 2017).

Finally, many libraries include some tools for visualizing and plotting results. Notably coach (Caspi et al., 2017) provides an interactive dashboard for exploring a variety of metrics which are automatically tracked during training.

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

# A   SUPPLEMENTARY MATERIALS

## A.1   ALGORITHM ACRONYMS

The expanded acronymns for all of the algorithms listed in Table 3 are given below:

- A2C: Advantage Actor-Critic
- ACER: Actor-Critic with Experience Replay
- ACKTR: Actor-Critic using Kronecker-Factored Trust Region
- Ape-X: Distributed Prioritized Experience Replay
- ARS: Augmented Random Search
- C51: Categorical DQN
- CAT-DQN: Categorical DQN
- CRA: Categorical Return Approximation
- CEM: Cross Entropy Method
- CER: Combined Experience Replay
- DDPG: Deep Deterministic Policy Gradients
- DQN: Deep Q Networks
- DQN+: Deep Q Network modifications, including some or all of the following: Double DQN, Dueling DQN, Prioritized Experience Replay
- DQFD: Deep Q-Learning from Demonstrations
- ES: Evolutionary Strategies
- GAIL: Generative Adversarial Imitation learning
- HER: Hindsight Experience Replay
- IMPALA: Importance Weighted Actor-Learner Architecture
- IQN: Implicit Quantile Networks
- MCTS: Monte Carlo Tree Search
- NAF: Normalized Advantage Functions
- NEC: Neural Episodic Control
- NSQ: n-step Q-learning
- OC: Option-Critic
- PPO: Proximal Policy Optimization
- QR: Quantile Regression
- QR-DQN: Quantile Regression DQN
- TD3: Twin Delayed Deep Deterministic Policy Gradient
- TRPO: Trust Region Policy Optimization
- SAC: Soft Actor-Critic

## A.2   ATARI RESULTS

The table below presents results for 62 Atari games. All agents were trained for 10M frames (40M including skipped frames). Reported results are the episode score at the end of training, averaged over the previous 100 evaluation checkpoints with each checkpoint averaged over 4 Sessions. Agents were checkpointed every 10k training frames. A Random baseline with score averaged over 100 episodes is included.

| Environment | Algorithm | | | | | |
| --- | --- | --- | --- | --- | --- | --- |
| (Atari, Discrete) | Random | DQN | DDQN+PER | A2C (GAE) | A2C (n-step) | PPO |

| | | | | | |
|---|---|---|---|---|---|
| Adventure | -0.89 | -0.94 | -0.92 | -0.77 | -0.85 | **-0.3** |
| AirRaid | 486 | 1876 | 3974 | **4202** | 3557 | 4028 |
| Alien | 97.0 | 822 | 1574 | 1519 | **1627** | 1413 |
| Amidar | 1.8 | 90.95 | 431 | 577 | 418 | **795** |
| Assault | 308 | 1392 | 2567 | 3366 | 3312 | **3619** |
| Asterix | 307 | 1253 | **6866** | 5559 | 5223 | 6132 |
| Asteroids | 1331 | 439 | 426 | **2951** | 2147 | 2186 |
| Atlantis | 29473 | 68679 | 644810 | **2747371** | 2259733 | 2148077 |
| BankHeist | 13.4 | 131 | 623 | 855 | 1170 | **1183** |
| BattleZone | 3080 | 6564 | 6395 | 4336 | 4533 | **13649** |
| BeamRider | 355 | 2799 | **5870** | 2659 | 4139 | 4299 |
| Berzerk | 212 | 319 | 401 | **1073** | 763 | 860 |
| Bowling | 24.14 | 30.29 | **39.5** | 24.51 | 23.75 | 31.64 |
| Boxing | -0.91 | 72.11 | 90.98 | 1.57 | 1.26 | **96.53** |
| Breakout | 1.26 | 80.88 | 182 | 377 | 398 | **443** |
| Carnival | 905 | 4280 | **4773** | 2473 | 1827 | 4566 |
| Centipede | 2888 | 1899 | 2153 | 3909 | 4202 | **5003** |
| ChopperCommand | 735 | 1083 | **4020** | 3043 | 1280 | 3357 |
| CrazyClimber | 2452 | 46984 | 88814 | 106256 | 109998 | **116820** |
| Defender | 546810 | 281999 | 313018 | **665609** | 657823 | 534639 |
| DemonAttack | 346 | 1705 | 19856 | 23779 | 19615 | **121172** |
| DoubleDunk | -16.48 | -21.44 | -22.38 | **-5.15** | -13.3 | -6.01 |
| ElevatorAction | 9851 | 32.62 | 17.91 | **9966** | 8818 | 6471 |
| Enduro | 0.0 | 437 | 959 | 787 | 0.0 | **1926** |
| FishingDerby | -93.96 | -88.14 | -1.7 | 16.54 | 1.65 | **36.03** |
| Freeway | 0.0 | 24.46 | 30.49 | 30.97 | 0.0 | **32.11** |
| Frostbite | 68.0 | 98.8 | **2497** | 277 | 261 | 1062 |
| Gopher | 276 | 1095 | **7562** | 929 | 1545 | 2933 |
| Gravitar | 219 | 87.34 | 258 | 313 | **433** | 223 |
| Hero | 706 | 1051 | 12579 | 16502 | **19322** | 17412 |
| IceHockey | -9.87 | -14.96 | -14.24 | **-5.79** | -6.06 | -6.43 |
| Jamesbond | 13.0 | 44.87 | **702** | 521 | 453 | 561 |
| JourneyEscape | -18095 | -4818 | -2003 | **-921** | -2032 | -1094 |
| Kangaroo | 54.0 | 1965 | **8897** | 67.62 | 554 | 4989 |
| Krull | 1747 | 5522 | 6650 | 7785 | 6642 | **8477** |
| KungFuMaster | 865 | 2288 | 16547 | 31199 | 25554 | **34523** |
| MontezumaRevenge | 0.0 | 0.0 | 0.02 | 0.08 | 0.19 | **1.08** |
| MsPacman | 170 | 1175 | 2215 | 1965 | 2158 | **2350** |
| NameThisGame | 2088 | 3915 | 4474 | 5178 | 5795 | **6386** |
| Phoenix | 1324 | 2909 | 8179 | 16345 | 13586 | **30504** |
| Pitfall | -301 | -68.83 | -73.65 | -101 | **-31.13** | -35.93 |
| Pong | -20.4 | 18.48 | 20.5 | 19.31 | 19.56 | **20.58** |
| Pooyan | 515 | 1958 | 2741 | 2862 | 2531 | **6799** |
| PrivateEye | -731 | **784** | 303 | 93.22 | 78.07 | 50.12 |
| Qbert | 157 | 5494 | 11426 | 12405 | **13590** | 13460 |
| Riverraid | 1554 | 953 | **10492** | 8308 | 7565 | 9636 |
| RoadRunner | 35.0 | 15237 | 29047 | 30152 | 31030 | **32956** |
| Robotank | 1.78 | 3.43 | **9.05** | 2.98 | 2.27 | 2.27 |
| Seaquest | 106 | 1185 | **4405** | 1070 | 1684 | 1715 |
| Skiing | -17361 | -14094 | **-12883** | -19481 | -14234 | -24713 |
| Solaris | 2097 | 612 | 1396 | 2115 | **2236** | 1892 |
| SpaceInvaders | 164 | 451 | 670 | 733 | 750 | **797** |
| StarGunner | 645 | 3565 | 38238 | 44816 | 48410 | **60579** |
| Tennis | -24.0 | -23.78 | **-10.33** | -22.42 | -19.06 | -11.52 |
| TimePilot | 3151 | 2819 | 1884 | 3331 | 3440 | **4398** |
| Tutankham | 15.45 | 35.03 | 159 | 161 | 175 | **211** |
| UpNDown | 125 | 2043 | 11632 | 89769 | 18878 | **262208** |
| Venture | 0.0 | 4.56 | 9.61 | 0.0 | 0.0 | **11.84** |
| VideoPinball | 25365 | 8056 | **79730** | 35371 | 40423 | 58096 |

| | | | | | | |
|---|---|---|---|---|---|---|
| WizardOfWor | 784 | 869 | 328 | 1516 | 1247 | **4283** |
| YarsRevenge | 3369 | 5816 | 15698 | **27097** | 11742 | 10114 |
| Zaxxon | 0.0 | 442 | 54.28 | 64.72 | 24.7 | **641** |

## A.3 Learning curves for all algorithms and environments

Each learning curve depicts the episode score, averaged over the previous 100 evaluation checkpoints, with each checkpoint averaged over 4 Sessions. The shaded area depicts +/- one standard deviation calculated over 4 Sessions.

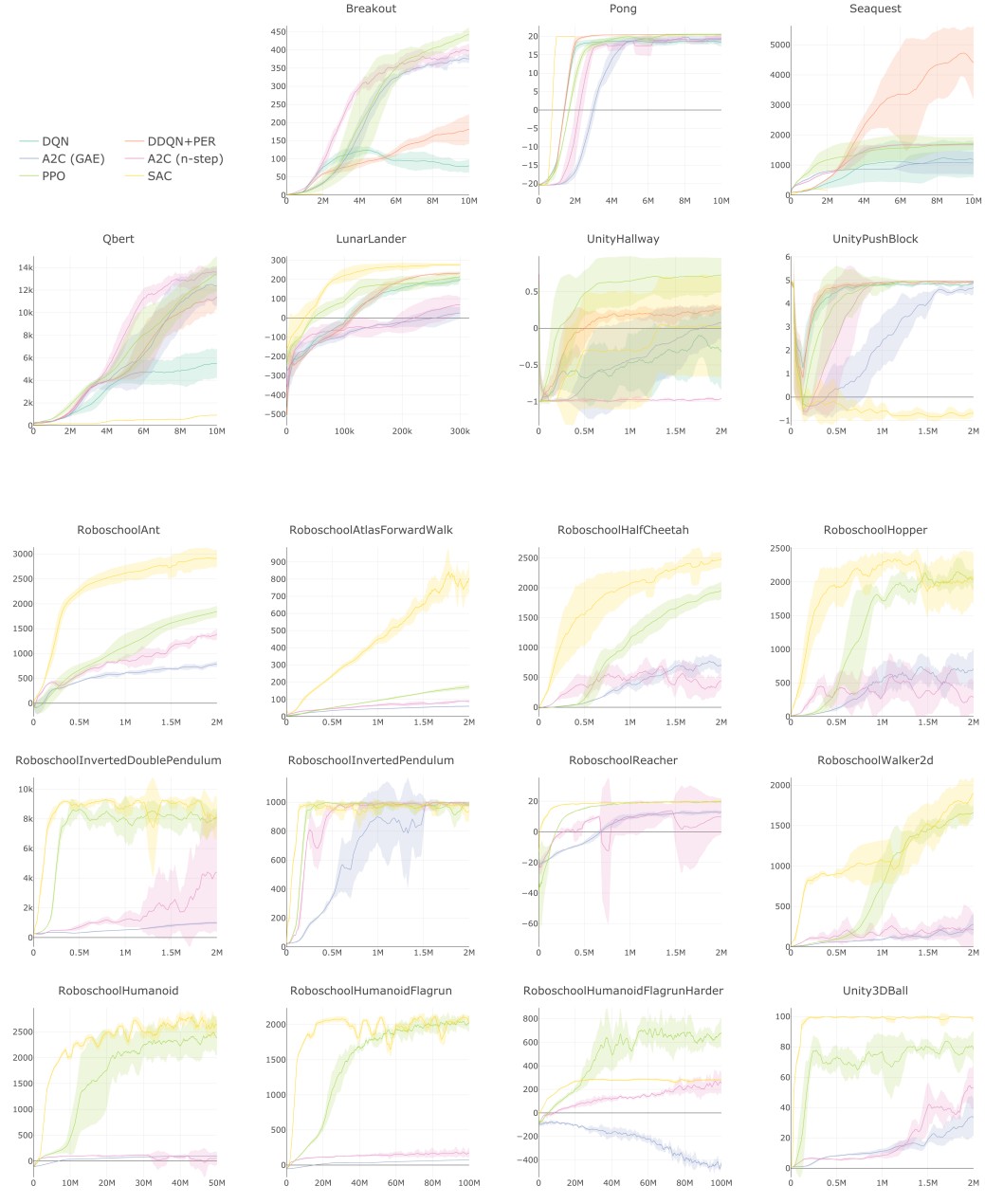

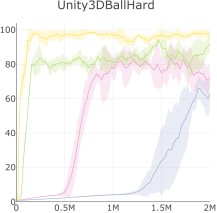

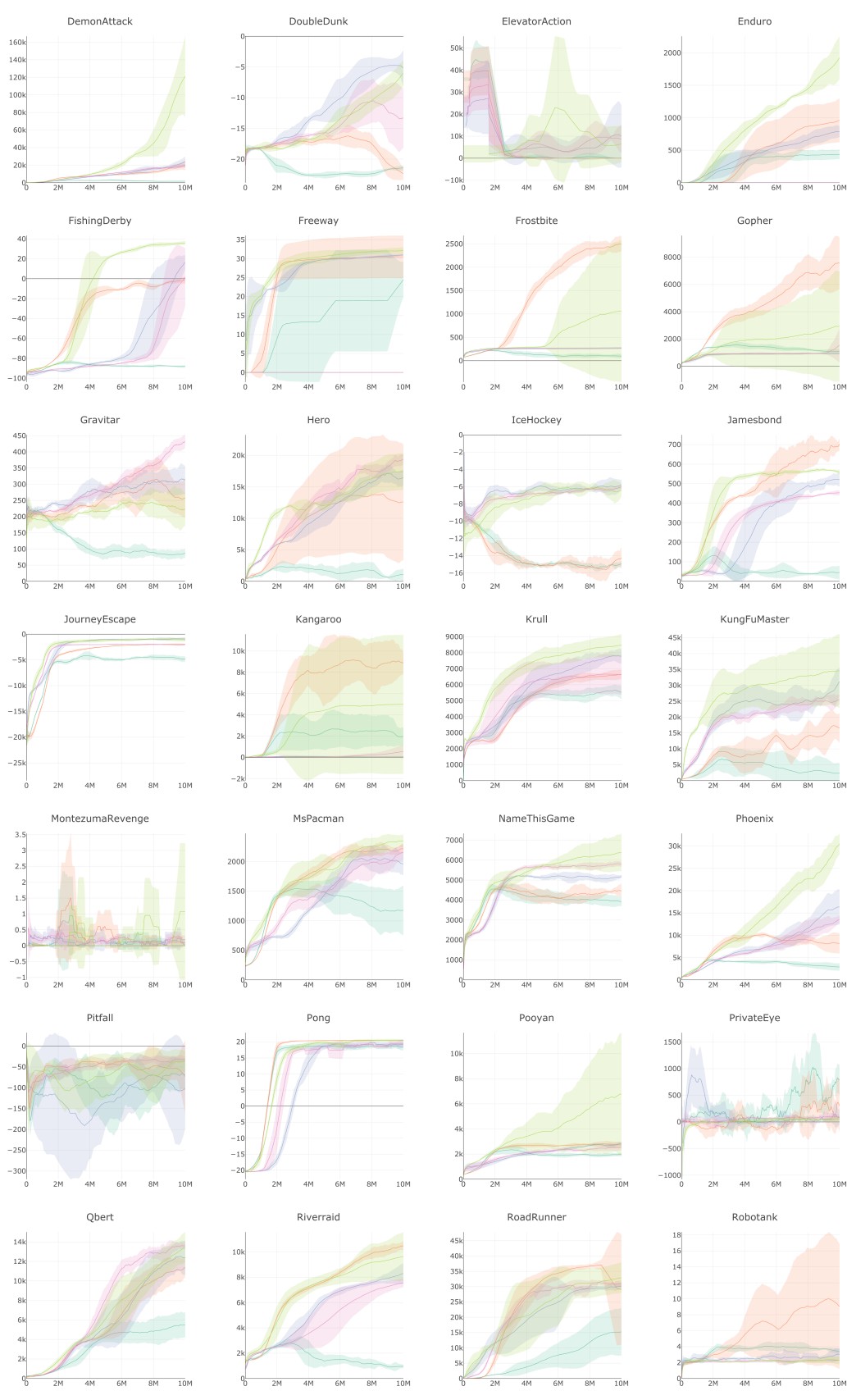

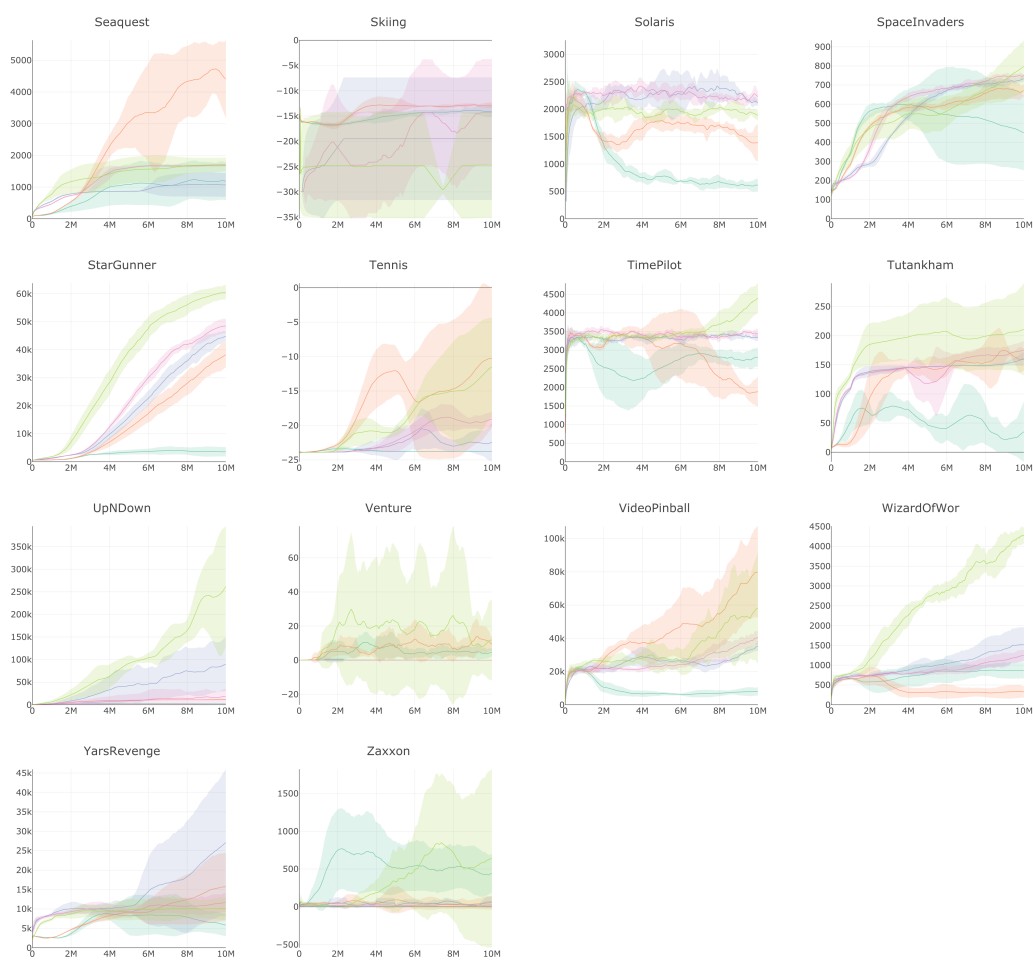

## A.4 EXAMPLE SPEC FILES

```
1  {
2    "ddqn_per_atari": {
3      "agent": [{
4        "name": "DoubleDQN",
5        "algorithm": {
6          "name": "DoubleDQN",
7          "action_pdtype": "Argmax",
8          "action_policy": "epsilon_greedy",
9          "explore_var_spec": {
10           "name": "linear_decay",
11           "start_val": 1.0,
12           "end_val": 0.01,
13           "start_step": 10000,
14           "end_step": 1000000
15         },
16         "gamma": 0.99,
17         "training_batch_iter": 1,
18         "training_iter": 4,
19         "training_frequency": 4,
20         "training_start_step": 10000
21       },
22       "memory": {
23         "name": "PrioritizedReplay",
24         "alpha": 0.6,
```

```
25        "epsilon": 0.0001,
26        "batch_size": 32,
27        "max_size": 200000,
28        "use_cer": false,
29      },
30      "net": {
31        "type": "ConvNet",
32        "conv_hid_layers": [
33          [32, 8, 4, 0, 1],
34          [64, 4, 2, 0, 1],
35          [64, 3, 1, 0, 1]
36        ],
37        "fc_hid_layers": [256],
38        "hid_layers_activation": "relu",
39        "init_fn": null,
40        "batch_norm": false,
41        "clip_grad_val": 10.0,
42        "loss_spec": {
43          "name": "SmoothL1Loss"
44        },
45        "optim_spec": {
46          "name": "Adam",
47          "lr": 2.5e-5,
48        },
49        "lr_scheduler_spec": null,
50        "update_type": "replace",
51        "update_frequency": 1000,
52        "gpu": true
53      }
54    }],
55    "env": [{
56      "name": "${env}",
57      "frame_op": "concat",
58      "frame_op_len": 4,
59      "reward_scale": "sign",
60      "num_envs": 16,
61      "max_t": null,
62      "max_frame": 1e7
63    }],
64    "body": {
65      "product": "outer",
66      "num": 1
67    },
68    "meta": {
69      "distributed": false,
70      "eval_frequency": 10000,
71      "log_frequency": 10000,
72      "rigorous_eval": 0,
73      "max_session": 4,
74      "max_trial": 1
75    },
76    "spec_params": {
77      "env": [
78        "BreakoutNoFrameskip-v4", "PongNoFrameskip-v4", "QbertNoFrameskip
    -v4", "SeaquestNoFrameskip-v4"
79      ]
80    }
81  },
82 }
```

Listing 1: The Double DQN + PER spec file for Atari games

```
1 {
2   "ppo_atari": {
3     "agent": [{
```

```
 4          "name": "PPO",
 5          "algorithm": {
 6            "name": "PPO",
 7            "action_pdtype": "default",
 8            "action_policy": "default",
 9            "explore_var_spec": null,
10            "gamma": 0.99,
11            "lam": 0.70,
12            "clip_eps_spec": {
13              "name": "no_decay",
14              "start_val": 0.10,
15              "end_val": 0.10,
16              "start_step": 0,
17              "end_step": 0
18            },
19            "entropy_coef_spec": {
20              "name": "no_decay",
21              "start_val": 0.01,
22              "end_val": 0.01,
23              "start_step": 0,
24              "end_step": 0
25            },
26            "val_loss_coef": 0.5,
27            "time_horizon": 128,
28            "minibatch_size": 256,
29            "training_epoch": 4
30          },
31          "memory": {
32            "name": "OnPolicyBatchReplay",
33          },
34          "net": {
35            "type": "ConvNet",
36            "shared": true,
37            "conv_hid_layers": [
38              [32, 8, 4, 0, 1],
39              [64, 4, 2, 0, 1],
40              [32, 3, 1, 0, 1]
41            ],
42            "fc_hid_layers": [512],
43            "hid_layers_activation": "relu",
44            "init_fn": "orthogonal_",
45            "normalize": true,
46            "batch_norm": false,
47            "clip_grad_val": 0.5,
48            "use_same_optim": false,
49            "loss_spec": {
50              "name": "MSELoss"
51            },
52            "actor_optim_spec": {
53              "name": "Adam",
54              "lr": 2.5e-4,
55            },
56            "critic_optim_spec": {
57              "name": "Adam",
58              "lr": 2.5e-4,
59            },
60            "lr_scheduler_spec": {
61              "name": "LinearToZero",
62              "frame": 1e7
63            },
64            "gpu": true
65          }
66        }],
67        "env": [{
68          "name": "${env}",
```

```
69        "frame_op": "concat",
70        "frame_op_len": 4,
71        "reward_scale": "sign",
72        "num_envs": 16,
73        "max_t": null,
74        "max_frame": 1e7
75      }],
76      "body": {
77        "product": "outer",
78        "num": 1
79      },
80      "meta": {
81        "distributed": false,
82        "eval_frequency": 10000,
83        "log_frequency": 10000,
84        "rigorous_eval": 0,
85        "max_session": 4,
86        "max_trial": 1
87      },
88      "spec_params": {
89        "env": [
90          "BreakoutNoFrameskip-v4", "PongNoFrameskip-v4", "QbertNoFrameskip
    -v4", "SeaquestNoFrameskip-v4"
91        ]
92      }
93    },
94 }
```

Listing 2: The PPO spec file for Atari games

```
1 {
2    "ppo_roboschool": {
3      "agent": [{
4        "name": "PPO",
5        "algorithm": {
6          "name": "PPO",
7          "action_pdtype": "default",
8          "action_policy": "default",
9          "explore_var_spec": null,
10         "gamma": 0.99,
11         "lam": 0.95,
12         "clip_eps_spec": {
13           "name": "no_decay",
14           "start_val": 0.20,
15           "end_val": 0.20,
16           "start_step": 0,
17           "end_step": 0
18         },
19         "entropy_coef_spec": {
20           "name": "no_decay",
21           "start_val": 0.0,
22           "end_val": 0.0,
23           "start_step": 0,
24           "end_step": 0
25         },
26         "val_loss_coef": 1.0,
27         "time_horizon": 2048,
28         "minibatch_size": 128,
29         "training_epoch": 10
30       },
31       "memory": {
32         "name": "OnPolicyBatchReplay",
33       },
34       "net": {
35         "type": "MLPNet",
```

```
36           "shared": false,
37           "hid_layers": [256, 256],
38           "hid_layers_activation": "relu",
39           "init_fn": "orthogonal_",
40           "clip_grad_val": 0.5,
41           "use_same_optim": false,
42           "loss_spec": {
43             "name": "MSELoss"
44           },
45           "actor_optim_spec": {
46             "name": "Adam",
47             "lr": 3e-4,
48           },
49           "critic_optim_spec": {
50             "name": "Adam",
51             "lr": 3e-4,
52           },
53           "lr_scheduler_spec": null,
54           "gpu": false
55         }
56       }],
57       "env": [{
58         "name": "${env}",
59         "num_envs": 8,
60         "max_t": null,
61         "max_frame": 2e6
62       }],
63       "body": {
64         "product": "outer",
65         "num": 1
66       },
67       "meta": {
68         "distributed": false,
69         "log_frequency": 1000,
70         "eval_frequency": 1000,
71         "rigorous_eval": 0,
72         "max_session": 4,
73         "max_trial": 1
74       },
75       "spec_params": {
76         "env": [
77           "RoboschoolAnt-v1", "RoboschoolAtlasForwardWalk-v1", "
      RoboschoolHalfCheetah-v1", "RoboschoolHopper-v1", "
      RoboschoolInvertedDoublePendulum-v1", "RoboschoolInvertedPendulum-v1
      ", "RoboschoolInvertedPendulumSwingup-v1", "RoboschoolReacher-v1", "
      RoboschoolWalker2d-v1"
78         ]
79       }
80   },
81 }
```

Listing 3: The PPO spec file for Roboschool environments (excluding Humanoid)

## A.5    KEY IMPLEMENTATION LESSONS

Implementing a number of well-known deep reinforcement learning algorithms taught us many important lessons. Such lessons are typically relegated to blog posts and personal conversations, but we feel it is worth sharing such knowledge in the research literature.

- Put *everything* that varies into the spec file. e.g. when implementing A2C we hardcoded state normalization in some places but not others. This extended the debugging process by days.

- Log policy entropy, $Q$ or $V$ function outputs and check that they change. Looking at $Q$ function values for a random agent helped us to track down an error due to state mutation.

- As with all Python code, watch out for mutation. For example, we noticed that the state returned from OpenAI baselines vector environment was mutated by preprocessing for the next time steps. This was because the state was tracked as internal variables and returned directly. To remedy this, the state must be returned as a copy.

- Start simple - implement parent algorithms first. e.g. REINFORCE, then A2C, then PPO or SAC, and ensure that each of these work in turn. PPO and SAC were straightforward to debug once A2C was working.

- Write tests: Tests have been invaluable in debugging and growing SLM Lab. Especially useful are tests for tricky functions. For example, a test for the GAE calculation was essential for debugging A2C. Additionally, use Continuous Integration to automatically build and test every new code changes in a Pull Request, as is done in SLM Lab.

- Check input and output tensor shapes for important calculations. e.g. predicted and target values for Q or V. Broadcasting errors may lead to incorrect tensor shapes which fail silently.

- Check the size of the loss at the beginning of training. Is this comparable to other implementations? Tracking this value was essential when debugging DQN. It led us to a bug in the image permutation and we also stopped normalizing the image values as a result.

- Check that the main components of the computation graph are connected as expected. This can be implemented as a decorator around the training step and run when debugging. Conversely, be careful to detach target values.

- Visualize trained agents. We forgot to evaluate agents with reward clipping turned off, which led to very significant underestimation of performance for all environments except Pong. We finally realized training was working only after deciding to look at some policies playing the game.

