# OpenReview forum: "SLM Lab: A Comprehensive Benchmark and Modular Software Framework for Reproducible Deep Reinforcement Learning"
_ICLR.cc/2020/Conference — Reject_

### Official Review · AnonReviewer2 · 2019-10-23
**Official Blind Review #2**

**Rating:** 8

**Review:**

SLM Lab is a software framework for reinforcement learning, which includes many different algorithms, networks, and memory types. The framework is well structured and modular. Thus, it is easily extendable for anyone and can be a pinnacle for future RL research.

The really like the paper. It is well written, easy to read, and provide a valuable platform / framework to the community, both the scientific community as well as practitioners. Although the scientific contribution may be low in the paper, I think the significance and potential impact of the paper outweigh that.

The paper also include many results from running the framework in various configurations, showing the flexibility and usefulness of it.

The code for SLM Lab is released open source, which is very valuable and enables future research in RL.


**Experience Assessment:**

I have read many papers in this area.

**Review Assessment: Checking Correctness Of Derivations And Theory:**

N/A

**Review Assessment: Checking Correctness Of Experiments:**

I assessed the sensibility of the experiments.

**Review Assessment: Thoroughness In Paper Reading:**

N/A

---

> ### Author Response · Authors · 2019-11-13
> **Author response to Official Blind Review #2**
>
> Thank you for taking the time to read the paper and for your comments.

---

### Official Review · AnonReviewer1 · 2019-10-24
**Official Blind Review #1**

**Rating:** 3

**Review:**

This paper presents a new RL library called « SLM Lab ». Its most relevant features for RL research are: (1) modularity to help re-use existing components (thus reducing the risk of subtle implementation differences when comparing algorithms), (2) implementations of most popular algorithms like DQN & variants, A3C, PPO, SAC, (3) ability to parallelize both actors (through vectorized environments) and the learner (through distributed gradient descent), and (4) utilities for hyper-parameter optimization, reproducible experiments and reporting. The paper also reports performance over Atari games, Roboschool environments as well as some Unity ML-Agents tasks. Finally, it provides a high-level overview of SLM Lab’s capabilities compared to 23 other RL open source libraries, showing that it is the only one that combines: reporting of the performance of the implemented algorithms, ability to specify hyper-parameters in the config file, parallelization, hyper-parameter optimization, and visualization of the results.

Overall this looks like a solid RL library, but I am not convinced that it brings enough novelty to the RL software landscape for a published ICLR paper — it would better fit in a workhop dedicated to ML libraries for instance, thus the weak reject.

Table 3 shows that SLM Lab is the only RL library with such a broad offering of features, and this is definitely impressive, but I would argue that many of these features can typically be added to other libraries by plugging in other open source software. For instance there are several tools for experiment management and hyper-parameter optimization (and for RLLib in particular, hyper-parameter optimization is not checked but is straightforward with Ray Tune). TensorBoard can also often be easily used for visualization.

The parallelization capabilities of SLM Lab also seem limited: if I understand correctly, actor parallelization can only occur on a single machine, and thus an algorithm like Ape-X or R2D2 could not be implemented. If this is correct then it is a major limitation of the framework, since such parallelization across multiple computers can be extremely useful when environments are slow and costly to run (in CPU / RAM).

It is not clear to me to which extent multi-agent is supported. It seems like it is possible to have multiple agents in one environment, but is that enough for general multi-agent RL? (ex: how to specify individual / team rewards? share information between agents? deal with agents not acting all at the same timestep? centralize part of the training / execution?…)

I appreciate that the paper presents a discrete version of SAC, mentioning how easy it was to implement thanks to the modular design of SLM Lab, but results from Table 1 do not look very good (especially since it did not work on some of the environments). Relying on the Gumbel-softmax might not be the most robust & stable way to train a discrete SAC — see e.g. the recent « Soft Actor-Critic for Discrete Action Settings » for a different approach.

Finally, it is also great to have some benchmarks of the algorithms being implemented, but at least for Atari, I am not aware of previous work using the exact same evaluation setting, so it is hard to tell how they compare to other implementations.

In spite of the above, I do not mean to criticize SLM Lab too heavily as from what I can tell it seems to be a a solid library with many useful features, and I am sure many researchers will find it useful in their day-to-day work.

Minor points:
- Anonymity was clearly violated with the two github links
- Having some synthetic result on Atari (like the typically reported median human-normalized score) would be good
- I am personally not a fan of large config JSON files due to the lack of comments in JSON
- A.5 (« Key Implementation Lessons ») is great!

Review update after author feedback: I am on the fence for this paper, but still leaning towards rejection due to the fact I am still not convinced that this library brings that much novelty compared to existing other libraries (although it seems like a nice RL library, I am not sure ICLR is the right venue for talking about it). The authors argue that their benchmark results are a key contribution of the paper, but I do not find these results particularly insightful, especially the Atari ones that are not comparable to previous results (due to using a different evaluation method) and the lack of state-of-the-art algorithms like Rainbow or IQN.

**Experience Assessment:**

I have read many papers in this area.

**Review Assessment: Checking Correctness Of Derivations And Theory:**

N/A

**Review Assessment: Checking Correctness Of Experiments:**

I assessed the sensibility of the experiments.

**Review Assessment: Thoroughness In Paper Reading:**

I read the paper thoroughly.

---

> ### Author Response · Authors · 2019-11-13
> **Author response to Official Blind Review #1**
>
> Thank you for such a comprehensive and thorough review, we really appreciate your comments. Please see our replies below.
>
> ---
> “...I am not convinced that it brings enough novelty to the RL software landscape…”
>
> SLM Lab presents an empirical contribution and software aimed to address reproducibility problems in RL. We think  this paper makes the following novel contributions to RL research:
>
> 1.1) Fair comparison of policy gradient and value-based methods from a single codebase with minimal implementation differences. This is the most extensive comparison we are aware of among the RL libraries we listed.
>
> 1.2) Hybrid parallelization: As far as we are aware, none of the RL libraries we listed provides this capability. An RL algorithm can be bottlenecked by stepping the environment or updating the networks, and this method is useful for finding the right mix of parallelization schemes to speedup training.
>
> 1.3) Method to address reproducibility in RL: We propose that RL libraries would benefit from having a spec file design which exposes all of the hyperparameters. Although not a new idea, the extent to which SLM Lab implements it is novel, e.g. by including the environment details, the hyperparameter search, and automatically savign the git SHA.
>
> ---
> “...many of these features can typically be added to other libraries by plugging in other open source software…”
> We agree, and in fact SLM Lab also uses Ray Tune for hyperparameter optimization and Tensorboard for visualization.   However, integrating other software libraries into a framework to function correctly takes significant time and effort, and our goal is to take that burden away from the users and let them focus on research.
>
> ---
> “The parallelization capabilities of SLM Lab also seem limited…parallelization can only occur on a single machine...”
> Indeed, SLM Lab can only parallelize on a single machine, but its parallelization capabilities are unique. It allows for hybrid parallelization: on the environment and on network training. We document the benefits of this novel contribution in the paper.
>
> ---
> “It is not clear to me to which extent multi-agent is supported….”
> Multi-agent is on our future roadmap, and the current spec file design is designed for future format compatibility with multi-agent.
>
> ---
> “...the paper presents a discrete version of SAC, … but results from Table 1 do not look very good…”
> Discrete SAC on Pong was the most sample efficient of all the algorithms, roughly by 2x compared to the next most sample-efficient algorithm. We included new discrete SAC results in the learning curve in the Appendix and Table 1.
>
> The other discrete SAC results are indeed not strong, but we felt it important to include also negative results, especially since the Pong result shows it is possible for discrete SAC to perform very well on vision-based environments. However, we have been unable to obtain good results uniformly across the Atari games. We feel this is useful for the research community to know.
>
> The “Soft Actor-Critic for Discrete Action Settings” you mentioned was released after our submission, and it is difficult to compare with their results because they focused on very few samples (100k frames). Their reported results are lower than ours in Table 1, and some of their results are worse than random.
>
> ---
> “...I am not aware of previous work using the exact same evaluation setting, so it is hard to tell how they compare to other implementations….”
>
> The difficulty you cite of comparing results across algorithms was one of the main motivations for SLM Lab. In addition to the variability in performance across different implementations, evaluation techniques in deep RL have changed over time.
>
> For example, some papers select the best policy during training (e.g. Prioritized Experience Replay, Schaul et. al, 2016) whereas others report results on the final policy (e.g. The Arcade Learning Environment: An Evaluation Platform for General Agents, Bellemare et. al., 2013).
>
> Our approach to evaluation for all environments is very similar to the proposal in “Revisiting the Arcade Learning Environment: Evaluation Protocols and Open Problems for General Agents”, Machado et. al, 2017 which recommends:
>
> “At the end of training (and ideally at other points as well) report the average performance of the last k episodes. This protocol does not use the explicit evaluation phase, thus requiring an agent to perform well while it is learning. This better aligns the performance metric with the goal of continual learning while also simplifying experimental methodology.”
>
> ---
> “Having some synthetic result on Atari … median human-normalized score…”
> Like most recent RL papers, we do not include human baseline scores since RL algorithms have exceeded human performance at Atari games, and they are also difficult to obtain. However, random baseline scores are easily generated in SLM Lab, so we have added a new “Random” column in the tables for better comparison.

---

### Official Review · AnonReviewer4 · 2019-10-31
**Official Blind Review #4**

**Rating:** 3

**Review:**

Summary:

The paper provides a description of a new framework for reproducible and efficient RL experiments, as well as benchmarks of many algorithms on popular environments, such as `Atari and Roboschool.

Pros:
- I agree that reproducibility is an extremely important question for the RL research, and thus such a code library is very beneficial for the community.
- The library is well designed, and allows for creating extensions rather easily in the future.
- Benchmarks are quite extensive and instructive.

Cons:
-  Comparison with the library [1] is missing (see also [2] for description and benchmarks). As both libraries are focused on reproducibility and flexible implementations of algorithms, such a comparison would support authors claims.
- I am not sure that ICLR is the right venue for such paper. Perhaps a more specialized conference of a workshop would be better.
- Anonymity violation

Questions:
- How difficult it is to implement distributional algorithms in your framework?
- What about different exploration strategies? (Boltzmann, epsilon-greedy, parameter noise etc.). I guess it should be quite easy to make it configurable as well


[1] https://github.com/catalyst-team/catalyst
[2] https://arxiv.org/pdf/1903.00027.pdf


**Experience Assessment:**

I have read many papers in this area.

**Review Assessment: Checking Correctness Of Derivations And Theory:**

N/A

**Review Assessment: Checking Correctness Of Experiments:**

I assessed the sensibility of the experiments.

**Review Assessment: Thoroughness In Paper Reading:**

I made a quick assessment of this paper.

---

> ### Author Response · Authors · 2019-11-13
> **Author response to Official Blind Review #4**
>
>
>
> Thank you for taking the time to review this paper and for your thoughtful comments. Please see our responses below.
>
> ---
> “Comparison with the library [1] is missing…”
> Thanks for bringing this library to our attention. We added a comparison in Table 3, and the similarities and differences are summarized below:
>
> Both libraries addresses reproducibility using config/spec files, although SLM Lab uses the git SHA to reference code as opposed to saving source as Catalyst does. They both report benchmark results, however SLM Lab is more comprehensive. Parallelization in Catalyst can scale to multiple machines, where as parallelization in SLM Lab is focused on the single machine use case. SLM Lab also logs to Tensorboard, and provides a more extensive automatic experiment analysis. Finally, Catalyst does not appear to provide hyper-parameter search. This is a key feature of SLM Lab and is configured in the spec file.
>
> ---
> “I am not sure that ICLR is the right venue for such paper...”
> Our paper is as much an empirical contribution as a software contribution. It provides what is, to the best of our knowledge, the most comprehensive set of benchmark RL results published to date (including entirely new results for the SAC algorithm), and moreover one which is a fairer comparison between RL algorithms than previous benchmarks due to the SLM Lab design that minimizes implementation differences. It also includes a new hybrid parallelization capability applicable to all RL algorithms.
>
> Even considering just the software contribution of the SLM Lab library, however, we feel that ICLR is an appropriate venue for this paper.  We note that the call for papers specifically lists “implementation issues, parallelization, software platforms, hardware” as a relevant topic. We also note that some of the libraries cited in Table 3 have been published at similar venues, such as ELF at NeurIPS 2017, and RLLib at ICML 2018.
>
> ---
> “How difficult it is to implement distributional algorithms in your framework?”
> This is on the SLM Lab roadmap.  It is not difficult, and can be implemented as an extension of the DQN class with a custom network output sampling mechanism and loss computation.
>
> ---
> “What about different exploration strategies?...”
> Boltzmann and epsilon-greedy exploration strategies are implemented and can be specified in the spec file. We have updated the paper to make it clear that this is available.
>
> Parameter noise is not currently implemented, but adding it is relatively straightforward, and it is on our future roadmap.

---

### Decision · Program_Chairs · 2019-12-19

**Decision:**

Reject

**Comment:**

A new software framework fo Deep RL is introduced. This is a useful work for the community, but it is not a research work. I agree with Reviewer4 that somehow it is not a right venue: other papers need to have technical contributions, SOTA, and here - it is difficult but it is another type of work - accurate technical implementation and commenting. I do not feel right to have as it a paper on ICLR.